# Transferrable Framework Based on Knowledge Graphs for Generating Explainable Results in Domain-Specific, Intelligent Information Retrieval

Hasan Abu-Rasheed *, Christian Weber , Johannes Zenkert , Mareike Dornhöfer and Madjid Fathi

Department of Electrical Engineering and Computer Science, Institute of Knowledge Based Systems and Knowledge Management, University of Siegen, 57076 Siegen, Germany; christian.weber@uni-siegen.de (C.W.); johannes.zenkert@uni-siegen.de (J.Z.); m.dornhoefer@uni-siegen.de (M.D.); fathi@informatik.uni-siegen.de (M.F.)
* Correspondence: hasan.abu.rasheed@uni-siegen.de

**Abstract:** In modern industrial systems, collected textual data accumulates over time, offering an important source of information for enhancing present and future industrial practices. Although many AI-based solutions have been developed in the literature for a domain-specific information retrieval (IR) from this data, the explainability of these systems was rarely investigated in such domain-specific environments. In addition to considering the domain requirements within an explainable intelligent IR, transferring the explainable IR algorithm to other domains remains an open-ended challenge. This is due to the high costs, which are associated with intensive customization and required knowledge modelling, when developing new explainable solutions for each industrial domain. In this article, we present a transferable framework for generating domain-specific explanations for intelligent IR systems. The aim of our work is to provide a comprehensive approach for constructing explainable IR and recommendation algorithms, which are capable of adopting to domain requirements and are usable in multiple domains at the same time. Our method utilizes knowledge graphs (KG) for modeling the domain knowledge. The KG provides a solid foundation for developing intelligent IR solutions. Utilizing the same KG, we develop graph-based components for generating textual and visual explanations of the retrieved information, taking into account the domain requirements and supporting the transferability to other domain-specific environments, through the structured approach. The use of the KG resulted in minimum-to-zero adjustments when creating explanations for multiple intelligent IR algorithms in multiple domains. We test our method within two different use cases, a semiconductor manufacturing centered use case and a job-to-applicant matching one. Our quantitative results show a high capability of our approach to generate high-level explanations for the end users. In addition, the developed explanation components were highly adaptable to both industrial domains without sacrificing the overall accuracy of the intelligent IR algorithm. Furthermore, a qualitative user-study was conducted. We recorded a high level of acceptance from the users, who reported an enhanced overall experience with the explainable IR system.

**Keywords:** explainability; graph based XAI; information retrieval; knowledge graphs; domain-specific IR



## 1. Introduction

The fourth industrial revolution brought to light new concepts that utilize increasing amounts of industrial data. The important role of data is motivated by the potential of extractable knowledge hidden within it. Knowledge integration plays an important role in smart factories on a human, organizational, and technical level, where data is the central resource for extracting knowledge with the help of, e.g., data analytics or machine learning methods [1]. For this reason, modern decision-making and information-retrieval systems provide new methods for extracting this knowledge from data. This is intended to enable experts to use this knowledge for analysis, planning, and well-informed actions [2]. In this

context, the role of the expert in the decision-making processes is essential, since they are the ones who understand the implications of the extracted knowledge and can evaluate and integrate it further. However, the expert's ability to use the extracted data is influenced by their understanding of the knowledge-extraction methods [3]. This is because understanding the knowledge extraction algorithm helps the expert to be aware of its performance, accuracy, and limitations, which in turn reflects on their trust levels towards the automatically extracted knowledge. The development of intelligent information retrieval systems mainly focuses on increasing the accuracy of retrieval. Here, we adopt the definition of an IR system by William and Baeza-Yates (1992) [4]: a system that is able to match a user query to data objects stored in a database. Those data objects are usually documents with semi-structured or unstructured textual content. In this article, we consider the information retrieval task as the method to respond directly to a user query or to retrieve information that other algorithms then use to respond to a user query. The former case takes the form of a search engine, while the latter can, for example, retrieve documents from a database for a recommender system, which then generates the recommendation to the user. Machine learning (ML), and especially deep learning (DL), models were able to solve prediction tasks in IR systems with high accuracy, provided an availability of sufficient datasets. However, intelligent information retrieval algorithms are usually considered black-box models, where the reasoning of their predictions is not clear. This has an influence on the level of user's trust and acceptance of the retrieved information [5]. This represents a particular limitation in critical domains, where experts require a clear understanding of the reasoning behind the prediction, in order to adopt it in making a final decision [6]. This is one of the main drivers to develop the so-called open-box algorithms, which offer the possibility to explain their reasoning to the user. The idea here is that the reasoning of the intelligent model can inform the reasoning of the human expert and therefore support the expert's decision. The importance of explaining the prediction of intelligent information retrieval algorithms grew rapidly in the recent years, as a part of the research on explainable artificial intelligence (XAI) [2,5,7]. Our proposed framework, therefore, aims to generate high-level explanations for intelligent IR systems. The explanations are meant to reflect the reasoning of the intelligent algorithms for the users, allowing them to better evaluate and utilize the retrieved information in their decisions.

Explaining intelligent IR algorithms depends on the domain of application, since the explanations themselves are meant to clarify the algorithm's logic to the domain experts. This means that explainability functions require tailoring towards their industrial applications, which are considered domain specific, since they have strict requirements for the decision-making process. In practice, tailoring IR and explainability solutions implies additional implementation costs for handling domain-specific requirements. Additional costs include: (1) labor costs associated with the developers, who manage the integration of domain-specific requirements in the IR and explainability algorithms, (2) labor costs associated with the domain experts, who provide the insights on the domain requirements, e.g., through interviews, and (3) the costs associated with the complexity of technical solution and subsequent costs of the methods it uses, e.g., the need for advanced processing power. Therefore, the technical complexity is a result of the domain requirements' complexity. In order to limit those costs, our proposed framework is designed to provide a transferrable structure for explainable intelligent IR solutions. It minimizes the needs for tailoring the IR and explainability algorithms to each domain of application by adopting a semantic representation of the domain's knowledge and requirements. Since the domain's requirements are integrated into the concept of knowledge representation, the same IR and explainability algorithms can be used in multiple domains. The cost of developing explainable IR systems will therefore considerably decrease by implementing the same IR and explainability algorithms in multiple domains. The balance between developing a cost-effective solution and a domain-specific one is required for IR solutions; this is still a challenge for the current state of the art.

To address the explainability of intelligent IR systems and their transferability between multiple domains, we investigated the role of knowledge graphs in making black-box IR algorithms more interpretable, while providing the option to model domain requirements in a transferrable manner. We considered two research questions in this investigation:

1. How to generate domain-specific explanations for intelligent IR tasks.
2. How to enable transferring the same explainable IR algorithm to other domains, without compromising its performance.

We address these challenges by developing a framework that can link the IR and explainability algorithms to the domain requirements through a knowledge graph as a unified knowledge representation structure. We aim to integrate those requirements in the knowledge representation to: (1) allow the intelligent algorithms to utilize the domain's requirements inherently, since they are already integrated in the knowledge structure, and (2) provide a domain-agnostic semantic knowledge structure that both IR and explainability algorithms can use. The key concept in our approach that answers these questions is considering a semantic knowledge graph (KG) representation as a single source of truth (SSoT). This way, multiple information sources are fused in the graph, and called by the intelligent IR and explainability algorithms, to provide the user with understandable search results. Our proposed solution serves as a foundation, which can be used to develop explainable, domain flexible, IR systems. We use the term "domain flexible" to reflect the ability of the framework of being domain-agnostic by being transferrable between multiple domains, while being able to represent each domain's requirements in the explainable IR algorithm.

Among multiple knowledge representation structures, we choose knowledge graphs in our framework for the following reasons:

1. The ability of knowledge graph to contextualize the entities within the graph. Contextualizing the entity is a result of its semantic relations to other entities in the same graph. This feature provides more reliability to the IR algorithm and more meaningfulness of the generated explanations.
2. Unlike relational databases, the knowledge graph functions as a graph database. This means that graph theory methods can be directly implemented on the entities and relations in the knowledge graph. Graph databases enable an efficient querying of the elements as a graph and the use of basic statistics, such as the degree of elements. Furthermore, more complex statements, which are captured in the literature under the term "centrality measures" can be calculated efficiently for a future algorithmic extension of the presented approach. This enables, for example, considering more complex measures of connectivity to evaluate the retrieved information in the context of other information. Utilizing those methods from graph theory enhances the relatedness of retrieved results to a certain user query.
3. Knowledge graphs provide paths between any pair of entities. Those paths consist of in-between nodes and relations that connect those entities. Utilizing the concept of graph walks, i.e., navigating those graph paths following a predefined set of rules, the IR algorithm can identify more relevant results to the user query. Moreover, the graph path represents the reasoning of connecting two entities and thus explains the retrieval of one entity based on the others.

The proposed approach provides the methods to implement domain and expert requirements in the explainable IR algorithm. This in turn enables the system to be applicable in different domains through integrating requirements-engineering concepts explicitly into the process of model creation. The knowledge graph links domain requirements, explanations, and the domain data together. During the construction of the KG, information from domain requirements, expert knowledge, and textual data sources are taken into consideration and modelled through graph nodes and relations. The result is then used to generate explanations on how the intelligent IR algorithm is retrieving search results from the KG. The embedded knowledge in the KG relations allows one to overcome the

challenge of explaining a black-box IR algorithm [2,8,9]. It provides sufficient information to generate an understandable explanation that reflects the reasoning behind the IR-model's prediction. Explanations from the KG can be constructed visually or verbally, making them more human-understandable for the end-user.

In order to generate a transferrable solution, the construction of the knowledge graph is designed to capture high-level domain requirements, from the domain experts and from the data sources themselves. Once those requirements are defined, they are embedded into the KG uniformly. Then, they are retrieved with the same intelligent IR algorithm, since the IR uses the same KG structure, and not the raw data source, to retrieve information. This aspect of our framework provides a novel approach to address the challenge of creating a transferrable IR solution that is domain specific at the same time.

Our contributions in this article are summarized as follows:

1. The development of a comprehensive framework, with specialized components that are capable of integrating domain-specific requirements in the chosen semantic knowledge representation.
2. Developing a novel system structure that enables intelligent IR and explainability algorithms to considering domain requirements inherently, using the KG-based SSoT.

The development of a novel transferability approach minimizes the implementation costs by enabling the use of the same IR and explainability algorithms in multiple domains, without compromising their domain-specific requirements.

We tested the transferability of the proposed method within two different application domains: (1) semiconductor industry and (2) job recommendation. Our results show high user acceptance and IR accuracy in both domains, with minimal-to-zero changes in the KG construction process, the intelligent IR algorithm, and the generation of visual and textual explanations.

In the following sections of the article, we review the related literature in Section 2. Section 3 elaborates on the proposed transferrable, KG-based retrieval, and explainability framework. The evaluation use cases and metrics are demonstrated in Section 4 and the article is concluded in Section 5.

## 2. Related Work

Although the topic of explaining intelligent algorithms is not new [3,10], it has gained new perspectives since deep learning algorithms have become widespread [11]. This was due to the black-box nature of such algorithms, which challenged the ability to track the algorithms' predictions and understand their reasoning. Explaining AI algorithms followed several directions in the state of the art. Li et al., (2020) [2] classify these directions in two main categories: data-driven approaches, such as [12,13], and knowledge-aware ones [14,15]. Data-driven methods use the information from data and the intelligent model itself in generating a comprehensible interpretation of the model's behavior. Knowledge-aware methods use the explicit or implicit knowledge that can be extracted from the domain. This knowledge is used either to generate or to enhance the explanations of the intelligent IR or recommendation algorithms.

For generating explanations, knowledge modelling methods have been investigated in recent years to infer the reasoning behind retrieving certain results by the IR algorithm or recommending an item to a certain user [16,17]. Both data-driven and knowledge-aware methods, which are implemented in singular domains of application, supported explaining the intelligent prediction models. However, these approaches were also dependent on the models used for the information retrieval or recommendation tasks. Therefore, recent literature shows a high focus on developing model-agnostic algorithms, not only for the intelligent IR or recommendation tasks but also for generating explanations of their predictions. A model-agnostic explanation algorithm means that the explanations are generated independently of how the intelligent model works. The algorithm uses the input and the output of the intelligent model to generate explanations without relying on the model's internal behavior. An example of such model-agnostic, explainable recom-

mendation algorithms can be clearly seen in the work of Chen and Miyazaki (2020) [17]. They developed a task-specialized knowledge graph that serves as a general common knowledge source, to generate model-agnostic explanations of the intelligent recommendations. Their knowledge-aware approach uses the task-specialized graph to overcome the challenge of not having sufficient information from the databases to generate high-quality personalized explanations.

Knowledge graphs are knowledge modelling methods that support the generation of knowledge-aware explanations [2]. A KG is a network of interconnected entities, where each entity is represented by a graph node, and the relevance between the nodes is represented by relations [18]. Through the nodes and their relations, the KG is described as a set of triplets (head, relation, tail) as shown in Equation (1).

$$KG = \{(h, r, t) | h, t \in E, \ r \in R\} \tag{1}$$

where $h$ is the head entity, $t$ is the tail entity, and $r$ is the relation between them. $E$ represents the entity group, and $R$ is the relations group.

In the context of domain-specific textual documents, graph nodes are defined to represent the content of those documents, i.e., the words, sentences, or paragraphs [19]. Graph relations are then defined based on certain criteria, which should also reflect the domain's requirements. Alzhoubi [20] proposes the use of association rules mining (RAM) for enhancing graph construction from textual databases. The proposed approach extracts frequent subgraphs from the overall KG. Those subgraphs are then processed to produce feature vectors that represent the relations between nodes. A similar approach is used in [21], where vectors of the textual documents are considered as graph nodes, while the relations amongst them are calculated using the cosine similarity scores between document pairs.

The importance of model-agnostic solutions comes from their ability to use the same explanation concept and apply it for different intelligent models [11,17,22]. However, despite their independence of the intelligent models, explainability algorithms are still influenced by the domain of application [8]. Different domains require different content and shapes of explanations. Domain specifications also influence the intelligent IR or recommendation algorithms. An example of such domain-specific algorithms is found in the work of Naiseh (2020) [23], who proposes explainable design patterns for clinical decision support systems. Another example in the biomedical domain is in the work of Yang (2020) [6], who uses knowledge graphs to support the information retrieval task while considering the domain of application and its implications on the generated explanations. Their work builds on the flexibility that graph nodes and relations have, for embedding domain-related information while representing the data. Knowledge graph structures support the explainability of the system and serve as a foundation for the information retrieval itself, since they represent queryable graphical knowledge bases [6].

The specific nature of a domain can be captured from multiple sources, which include the databases and the knowledge of domain experts. Model-agnostic solutions, such as in [17], solve the challenge of explaining different black-box algorithms with the same explainability approach, but they do not handle components of the system that reflect its specific requirements, such as the role of domain experts in generating the knowledge graph, which then underlines the explainable recommendation algorithm. Domain-specific approaches such as [6,14,23] address the challenges of considering domain requirements. The compliance with domain requirements and specification(s) has its own cost, however. The more the explainable solution is tailored towards a specific domain, the less applicable it becomes in other domains of application [8]. For example, Ehsan et al. (2018) [14] describe a highly domain-specific task for an explainable robotic behavior. They take into consideration the expert's role in annotating the corpus and link it to the robot's navigation states. However, their explainable approach is still dependent on the intelligent model, meaning that other domains of application, with other requirements and databases, cannot use the same model. Their approach is applicable in other domains once new corpuses are

generated and the model is retrained again based on the new data. This process implies the costs we elaborated in Section 1.

Moon et al., (2019) approach a solution for both challenges, by developing a domain-agnostic intelligent reasoning system over graph walks for an entity retrieval task [11]. Their system is built on a conversational database (OpenDialKG) and tested amongst multiple domains. Their solution highlights the importance of bridging the gap between developing domain-specific solutions and transferring them amongst domains. Their solution shows high accuracy in adopting the same intelligent model to different domains. However, that adaptation is based solely on the large corpus they use to train the intelligent model. In a domain where limited data is available, the model may not be able to cover other domains. Moreover, the dependency on the data does not take into consideration the role of domain experts in modeling domain requirements.

Therefore, our proposed framework is designed to enable domain-specific, explainable algorithms, which are transferable to other domains with minimal changes. We put a high emphasis on domain modeling by including the role of domain experts and explicit domain requirements alongside the role of domain databases in the framework. Our approach allows generating high-level explanations that users can read in a natural language or visualize through the knowledge graph. Our solution builds on the work of Chen and Miyazaki (2020) [17] and Moon et al., (2019) [11] and bridges the gap between developing domain-agnostic and domain-specific explainable systems, utilizing the knowledge graphs as a base structure that is comprehensible and adaptable to different domains.

We propose a complete framework that considers the role of databases, domain experts, and domain requirements in constructing the knowledge graph and then furthering the IR explanations. Our framework uses the knowledge graph as the single source of truth, which provides the intelligent IR and recommendation algorithms with information. The explainability algorithm relies on the KG to generate high-level, textual, and visual explanations. In contrast to the approaches in [15,24,25], where the IR and recommendation algorithms are model-intrinsic, the use of the KG as a SSoT in our framework allows the explainable IR algorithm to be model-agnostic. This removes the limits on the IR and explainability algorithms alike. Moreover, as compared to the work of Moon et al. [11], our framework extends the sources of domain requirements. While their approach captures the requirements from databases solely, our framework enables capturing those requirements from databases, expert knowledge, and other requirement sources that can be explicitly defined, such as those found in the exploratory data analysis (EDA). Our proposed framework serves as a foundation for building domain-specific systems, since the knowledge graph is constructed with domain requirements in mind. Moreover, it addresses the transferability of the intelligent IR and explainability algorithms, through adopting the knowledge graph as the main source of information for both algorithms. In comparison to the similar transferrable approach of Chen and Miyazaki [17], our framework does not compromise the domain-specific nature of the IR solution when transferring it to other domains. We explicitly embed those requirements in the system design to enable their effective integration in the intelligent, explainable, IR. The use of the same intelligent models for information retrieval or recommendation in different domains is therefore possible, since they are trained to query the graph and not the data sources directly.

In Table 1, we summarize a part of the reviewed literature that informed our approach. The table compares between approaches for developing graph-based, explainable intelligent information retrieval and recommendation systems, including our proposed framework. We separate the features of the similar solutions from their final outcomes to focus the comparison on the two research questions of our research, namely addressing the domain-specific requirements in the IR solution and enabling the transferability of the solution to other domains with respect to their own sets of requirements.

**Table 1.** Comparison between different explainability approaches.

| Reference | Year | Model-Agnostic/ Intrinsic | Solution Features | | | Solution Outcomes | |
|---|---|---|---|---|---|---|---|
| | | | Recommendation Approach | Explainability Approach | Retrieval Task | Domain Requirement Based on | Transferable with Respect to Domain Requirements |
| Chen and Miyazaki [17] | 2020 | Model-agnostic | Conventional and intelligent recommendation systems | Textual explanation generated from a translator after ranking graph paths | Graph path retrieval | Not domain-specific | Yes |
| Moon et al. [11] | 2019 | Model-agnostic | Entity recommendations based on graph walker | Explanations are based on the generated paths from the graph walker | Graph path retrieval | The Database | Yes |
| Song et al. [24] | 2019 | Model-intrinsic | Based on a Markov decision process | Usage of user-to-item paths to generate explanations | Graph path retrieval in user-item-entity graph | Not domain specific | No |
| Wang et al. [15] | 2019 | Model-intrinsic | Intelligent model to learn path semantics and generate recommendations | Pooling algorithm to detect the path strength and its role in the prediction | Graph path retrieval | Not domain specific | No |
| Xie et al. [25] | 2021 | Model-intrinsic | Recommendation based on a user-item KG with item ratings | Explanation based on KG and multi-objective optimization | Graph path retrieval | Not domain specific | No |
| Our Framework | 2021 | Model-agnostic | Graph path recommendation | Graph based | Node retrieval, Graph path retrieval | Database, Expert Rules, Explicit requirements | Yes |

## 3. Transferable, Graph Based, Domain-Oriented, Explainable IR

In order to address the challenges of explaining intelligent IR algorithms while considering domain requirements, we use multiple components to build a transferrable structure that accomplishes this task. Our approach, visualized in Figure 1, is composed of domain-specific components and domain-agnostic ones. Domain-specific components represent the domain's data, requirements, and any rules defined by the domain's experts. Domain-agnostic components play two main roles in the proposed framework:

(1) They link domain-specific components to the knowledge graph, which enables a transferrable graph construction process that can be replicated for other domains.
(2) They perform the tasks of generating explanations and retrieving information based on the KG structure.

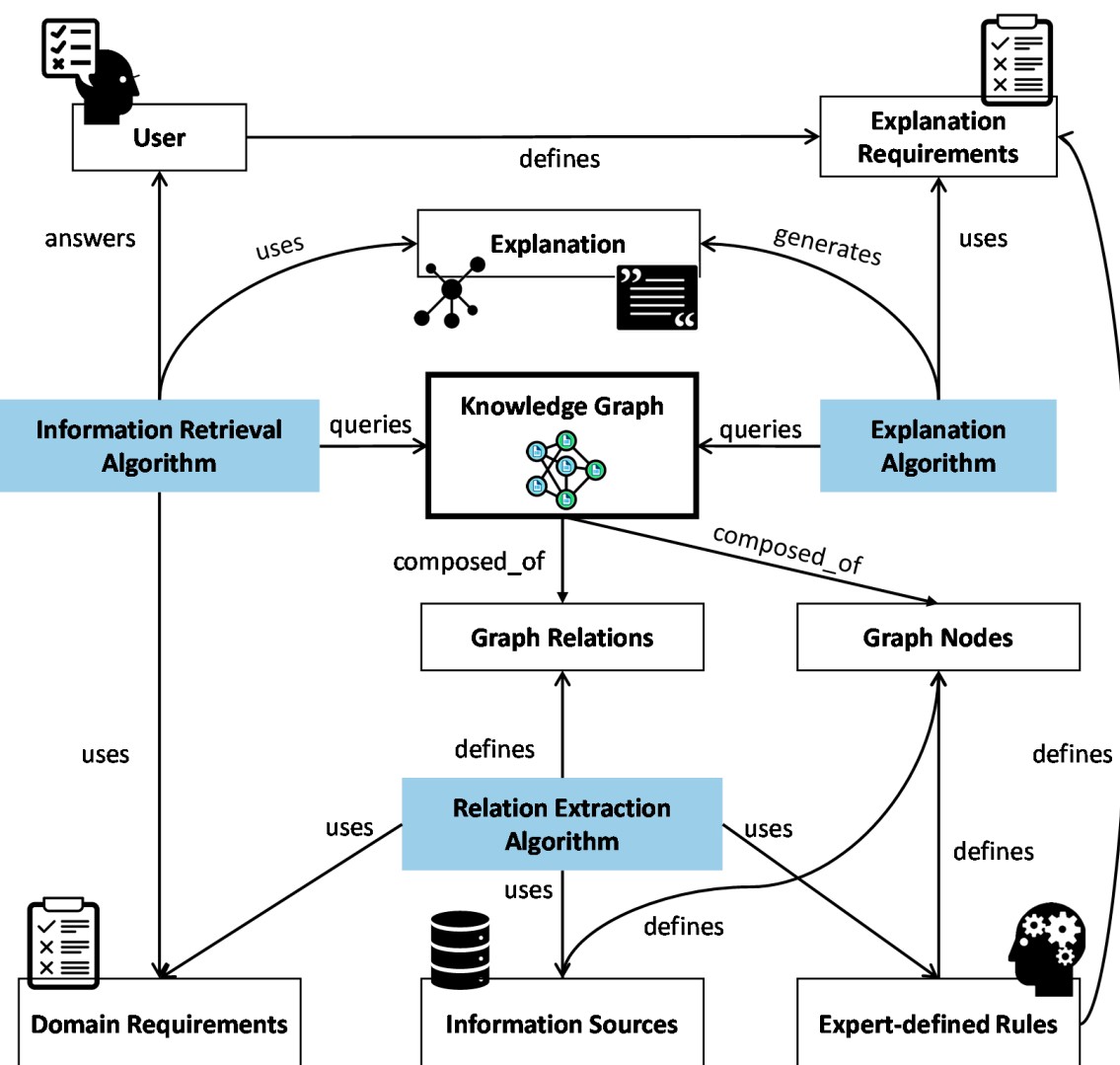

**Figure 1.** Proposed framework for transferrable, domain oriented, explainable IR, based on knowledge graphs. Domain-specific elements are shown in white, while domain-agnostic algorithms are shown in blue.

Domain-agnostic components integrate the intelligence of the IR algorithms, without compromising their explainability. This is due to the strong connection to the knowledge graph, which is inherently an open-box knowledge representation. The resulting framework that includes all previous components and links them together through the KG is highly flexible to the needs of multiple domains. Moreover, it enables flexible adaptation to multiple AI algorithms that are used in the information retrieval task, since they are dependent on the data that the KG itself is now providing.

The heart of our approach is the knowledge graph. The graph enables extracting information from multiple sources, which are databases, domain experts, and lists of requirements. It fuses this information in one queryable knowledge structure through defining the nodes and relations of the KG. Once the graph is constructed, it serves as the graphical knowledge base, which provides IR and explainability algorithms with all needed information to generate explainable results for the user. Since the explanation is generated based on the knowledge graph, our framework is capable of generating high-level explanations even for black-box IR algorithms. The explanation format, e.g., visual, textual, etc., is also dependent on the user's requirements. In the following sections, we explain the different parts of the framework in detail.

### 3.1. Knowledge Graph Construction

Constructing the knowledge graph is based on defining its nodes and relations. The knowledge graph is capable of representing multiple dimensions of a process. This is accomplished by: (1) defining node-types in the KG [21], (2) expanding the relation definition to connect different nodes from the same type or different types, while defining different levels of relevancies between the nodes [26]. In the proposed approach, graph nodes are defined based on two pillars: information sources and expert-defined rules. Each pillar participates in defining the node's type, as well as its content and attributes.

#### 3.1.1. Graph-Nodes Definition

Information in an industrial environment, for example, is usually distributed across multiple data sources. These sources can equally represent multiple processes or process parts. Therefore, our framework enables several information sources of defining different node types in the graph. Each node type represents the information source individually, while the graph relations connect those sources together. The node is then capable of representing the domain-specific nature of the information, without compromising the integration between multiple information sources in one knowledge structure.

Expert-defined rules are used to reflect the knowledge of domain experts on the construction of the knowledge graph. In the framework, we model the extracted expert knowledge through rules, because they allow a formal representation that supports making a decision about the node types, content, and the graph relations and explanations. An expert-defined rule has an "IF . . . THEN . . . " structure. Reasoning with these rules can follow a forward-chaining or a backward-changing approach [27–29].

There are no restrictions on the methods used to get the knowledge from domain experts. One can use questionnaires, interviews, and formal reports, for example, to extract expert knowledge. However, our experience in implementing this concept shows that conducting interviews with domain experts is an effective method for acquiring their knowledge and directly translating it into rules in the system. Then, a backward chaining inference allowed deducting the type and content of graph nodes, to represent the information sources. Figure 2 shows an example of an expert defined rule and its use in constructing the graph nodes.

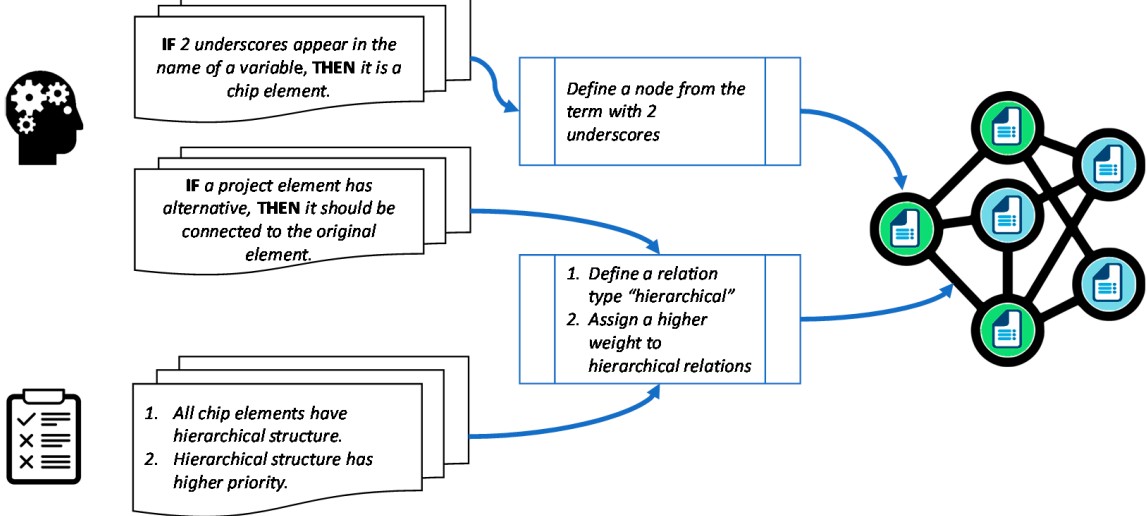

**Figure 2.** Example of integrating expert defined rules and domain requirements in the creation of graph nodes and relations. In the proposed framework, expert knowledge is used for defining the nodes and relations of the graph. Domain requirements are here presented as a list, which is used by the relation extraction algorithm to define graph relations.

### 3.1.2. Graph-Relations Extraction

Defining graph relations also uses the previous two pillars, i.e., expert-rules and data sources, and adds to them domain requirements. We use an independent domain-requirements block in our framework to represent domain-specific needs that cannot be directly represented in the information sources or the expert-defined rules. For example, the terminology used in the documentations differs from one domain to another. This terminology may also include certain entity types that have different meanings or implications in multiple domains. Information sources and expert-defined rules do not explicitly express those different implications. Therefore, we model such differences as domain requirements in the framework. During the exploratory data analysis (EDA), characteristics of the data reveal these implications, which are then translated to requirements that are considered within the graph construction, information retrieval, and explanation steps.

Domain requirements can take several formats. They can be represented by lists, dictionaries, or statements. They are then used within the relation extraction and the information retrieval steps. The developer of the IR algorithm can manually integrate those requirements into the algorithm in order to reflect them in graph queries. Figure 2 shows an example of using expert-defined rules and domain requirements to define graph nodes and relations.

We use a dedicated relation extraction (RE) algorithm to find the relations between the nodes. The relation extraction utilizes the type and content of each node in the graph, which reflects the content of the information sources. For example, nodes that represent documents have textual content. To extract the relations between those nodes, textual similarities are calculated between the node-pairs and added to the graph as a relation if the similarity score is higher than a pre-defined threshold. Graph relations can also be defined by the experts. In this case, the relation extraction algorithm translates the expert-defined rule into a relation between the corresponding nodes in the graph. An example of such a relation is linking document nodes based on a specific string of characters that appears in the document's header. Other domain requirements, which are not found during the EDA, are also used by the relation extraction algorithm to find links between the nodes, e.g., based on the implications of their domain terminology, Figure 2.

Through the information sources, expert-defined rules, and domain requirements, the relation extraction algorithm in our framework is informed by the domain's specific nature. The algorithm itself, however, is not limited by the domain. A developer can choose the relation extraction algorithm from a wide range of methods that are provided in the state of the art. This allows the same relation extraction algorithm to be used in multiple domains, while considering their domain-specific requirements at the same time. The combination of this domain-agnostic component with domain-specific components is what enables our approach to be transferred to other domains once their requirements and specific nature are known. This, in turn, is due to the use of the knowledge graph at the center of the framework.

### 3.2. Graph-Based Information Retrieval

Similar to the relation extraction algorithm, the information retrieval component in our framework is designed to be domain agnostic. This enables its re-use in multiple domains without compromising its ability to consider the domain-specific nature. Domain-specific information is provided to the IR algorithm through the knowledge graph, i.e., the definition of its nodes and relations. The information retrieval algorithm is also informed by the domain-requirements, since these requirements may affect graph queries. An intelligent IR generates predictions of correct responses to a user query. It can predict the nodes and relations in the knowledge graph that lead to the most relevant results. When it retrieves those results, it is also capable of getting explanations for each result from the explainability algorithm. The IR and explainability algorithms are related via the knowledge graph.

### 3.3. Graph-Based Explainability

To ensure the framework's ability of generating explanations for IR or recommendation algorithms, regardless of them being open-box or black-box ones, we separate the explainability algorithm in a way that only requires querying the KG to generate the explanations. To generate an explanation for a result using the KG, the algorithm determines the shortest path from a query node to possible, related result nodes in the KG [30]. The query node in this context represents the node in the KG, which is directly retrieved from the graph corresponding to a user query. Result nodes are the ones that are transitively related to the query nodes. Nodes that belong to a graph path between query and result nodes are ranked to identify the best fitting explanation for a retrieved result from the KG. To present the explanation in a high-level, user-friendly way, natural language processing (NLP) patterns are used to generate the output sentences.

The KG structure offers a visual explanation in addition to the textual one. Verbal explanation is achieved from the graph by re-phrasing the information as a human-readable sentence, e.g., through NLP and following the relevant nodes through relations. On the other hand, the graphical representation of the knowledge graph with its nodes and relations is also considered a visual explanation. Those visual explanations show node relevancies, node clusters, or dependencies. Multiple visual features, e.g., color or size, can be used to make the retrieved information more interpretable and understandable by the user. Figure 3 shows an example of generating those verbal and visual explanations from the KG.

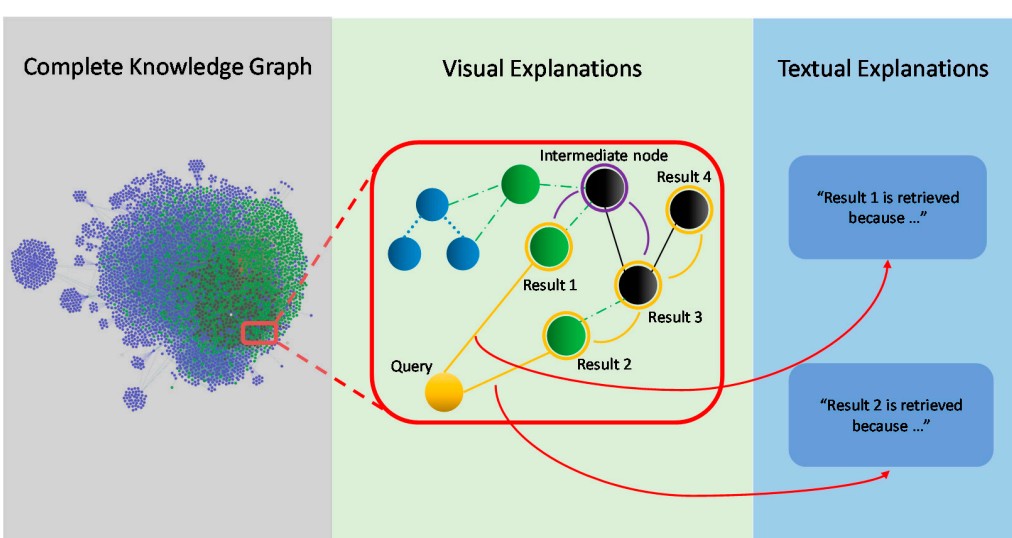

**Figure 3.** Visual and textual explanations based on the knowledge graph. Results that are retrieved by the IR algorithm are extracted from the graph and shown visually as a sub-graph, which shows the interlinks between different results. Textual explanations are generated from the reasoning behind linking two or more nodes in the graph to each other.

## 4. Experiment Design and Framework's Evaluation

In order to evaluate our framework, we designed an experimental setup to test the performance of each part of the framework. We focus on two main evaluation criteria, which correspond to our research questions:

1. The extent of the framework building a domain-specific solution that integrates the domain requirements and expert knowledge into the explainable, intelligent IR algorithm.
2. The transferability of the framework to different domains, without compromising the performance of the KG construction, the intelligent IR, and the explainability algorithms.

To evaluate we use a set of key performance indicators (KPIs). We use the KPIs to evaluate the performance of the framework in a certain domain, i.e., the first criteria. To evaluate the second criteria, we implement the framework on two different domains that include a high level of dependency on the domain requirements and the knowledge of domain experts. The domains belong to different environments to ensure that their requirements have no similarities. The first evaluation domain is supporting the semiconductor chip-design process. Within this domain, a use case of a semantic search for design documents has been implemented using the framework. The second evaluation domain is in the field of CV-Job matching and job-posting recommendations. The second use case takes the form of an intelligent recommender system for personalized job-posting suggestions.

In each of the two use cases, we tested (1) the performance of the intelligent information retrieval algorithm, (2) how the KG represents the domain of application, and (3) the explainability of the overall solution.

*Evaluation of the IR and Recommendation*

We tested the performance of the IR algorithm quantitatively through two KPIs: (1) the relevance of the semantic search results and (2) the precision and accuracy of the intelligent document clustering and matching. Moreover, we evaluate the algorithm qualitatively through expert-user feedback, which was collected through surveys and focus groups.

The *Relevance* measure, shown in Equation (2), is defined by the number of relevant results to a user's query, as a percentage of the total number of retrieved results.

$$Relevance = \frac{No.\ Relevant\ Results}{Total\ No.\ Results\ Retrieved} \times 100\% \tag{2}$$

The precision and accuracy are measured for the intelligent models that are implemented for the IR and recommendation tasks. We measure the precision and accuracy of those models using the standard *Precision* and $F_1$ measure, see Equations (3) and (4).

$$F_1 = 2 \times \frac{Precision \times Recall}{Precision + Recall} = \frac{\text{TP}}{\text{TP} + \frac{1}{2}(\text{FP} + \text{FN})} \tag{3}$$

$$Precision = \frac{\text{TP}}{\text{TP} + \text{FN}} \tag{4}$$

where TP is the number of correctly classified documents, while FP is the number of documents that are incorrectly classified as belonging to a certain class, and FN is the number of documents that are incorrectly classified as not belonging to that class.

*Evaluation of the Domain-Specific KG*

To test the knowledge graph's ability to represent the domain, we used an ontological structural-evaluation metric, namely the class richness [31,32], which is defined in Equation (5).

$$lass\ Richness = \frac{No.\ Classes\ With\ Instances}{Total\ No.\ Classes} \times 100\% \tag{5}$$

*Class Richness* measure reflects how the graph covers the instances of the use case. The underlying assumption for using this measure is that the elements of the framework, expect for the algorithms, can represent classes of an ontology. If a class is capable of representing a domain, then it will have instances defined from that domain. The more classes having instances, the more representative is the proposed framework of the domain of interest.

*Evaluation of the Explainability Algorithm*

Testing the explainability algorithm takes place through four different metrics. We measure firstly the availability of information for generating an explanation based on the knowledge graph. *Information availability* is used in this context to measure the quality of the explanation. It represents the amount of information that our framework provides

about the retrieved results from the KG. The more information collected about the algorithm's reasoning, the more comprehensive the explanation will be, and thus the higher the explanation quality is. To quantify this measure, we rely on the explanation templates that are used for generating the textual explanations. We consider the explanation to be of a high quality if all the slots of the explanation template can be filled with information from the KG. Otherwise, we consider the explanation to be of a low quality. We then define the *information availability* as the ratio of high-quality explanations to the total number of explanations generated by the explainability algorithm, as in Equation (6).

$$Information\ Availability = \frac{No.\ Fully\ Generated\ Explanations}{Total\ No.\ Explanations} \quad (6)$$

In addition to the information availability, we calculate the *mean explainability precision* (MEP) [17,33] of the explanation that are associated with the IR output, as defined in Equation (7). *MEP* measures the average proportion of explainable results of an IR algorithm or recommendations of a recommender system.

$$EP = \frac{1}{U} \sum_{u=1}^{U} \frac{N_{exp}}{L} \quad (7)$$

Here, $U$ is the number of users, $N_{exp}$ is the number of explainable results, and $L$ is the total number of results.

The third and fourth metrics focus on the quality of the explanation itself. We use *bilingual evaluation understudy* (BLEU) [34] and *Rouge-L* [35] scores to calculate how much a generated explanation represents the explained result. BLEU score measures the closeness between a machine-generated text and the original human-defined one. We use this score to measure how a generated textual explanation represents the original information that the KG paths have provided to the explainability algorithm. Rouge-L score measures the ability of a machine generated text to summarize the original information. In our case, this score represents the ability of the explanation to summarize the information that led to retrieving a certain result from the KG structure.

While $F_1$ measure, *Class Richness*, *MEP* score, and *BLEU* and *Rouge-L* scores are all well-defined measures that are used in the literature, we develop the *Relevance* and *Information Availability* metrics to correspond to the nature of our domain-oriented framework and to demonstrate its ability to perform accurate information retrieval and generate comprehensive explanations for the different domain-specific environments it is used for.

In the following, we elaborate the implementation of our framework within two evaluation use cases. We focus on the strategy of implementing the framework's components in each domain, the results of evaluating the framework's domain coverage, and its transferability.

*4.1. Use Case 1: Chip-Design Document Search and Retrieval*

The first use case for implementing and evaluating our framework is situated in the semiconductor industry. In the semiconductor chip-design environment, design documents include different types of reports, which are created during multiple design phases. The documents in our use case include three types, where each type is collected from a separate database:

1.  Failure reports: these represent the reports that design engineers prepare for each discovered failure in the chip design. The report is a semi-structured, text-based document. It contains predefined information fields, which are then filled with free textual input, describing the failure itself, its corrective actions, and additional relevant details.
2.  Project structures: this type of document is the result of the automated generation of a design project from the internal system of the company. It represents all project parts, product elements, and technical details, which have been inserted by the project

personnel. This data source inherits the structure of the design project itself. Such structural information is especially important for discovering links and relations between different elements of one project or, possibly, between different projects.

3.  Specifications documents: which also represent the full project documentation but in a human readable format, including tables of technical details, schematics of the chip modules, and so on. This type of file is considered as a reference, which a design engineer re-visits in case they need more information about a certain technical detail of the project.

These three document sources provide the foundation for lessons learned from previous chip designs. The goal of this use case is to extract and model this knowledge and to provide it to the design engineers with high-level explanations. When engineers are handling a certain design problem, they look for similar previous failures to see if their solutions can be used for the current problem. However, despite the availability of documents about previous failures, the engineer does not have sufficient tools to find semantically similar failures within large, multiple databases. Therefore, the information retrieval task in this use case takes the form of a semantic search engine, which is designed to provide the design engineer with relevant documents that are semantically similar to a current design failure. The engineer can write a short description of the current failure and search for relevant documents throughout the multiple document sources. The information retrieval algorithm searches the knowledge graph to provide the user with a list of search results. Each result includes a textual explanation of the reasoning behind its retrieval. Visual explanations are also provided to the user through visualizing a sub-graph that includes the search results, their interconnections, and their connection to the search query.

### 4.1.1. Dataset Preparation

Three datasets of design documents were used in this use case. The datasets included a total of 5587 documents, which were generated from design projects over a period of 10 years. Documents from the three datasets had different levels of structuring:

*   Documents with a hierarchical structure (Project structures): which include the project structure documentation.
*   Documents with a column-based labeled structure (Failure reports): which included the failure reports as a free textual input, arranged in semi-structured tables.
*   Unstructured documents (specification documents): which included a non-consistently structured content, describing the technical specifications of the final product.

We implemented a customized text mining pipeline to extract the textual content of the three document types. Extraction steps included cleaning the text from less meaningful terms, which can affect text weighing and similarity algorithms, such as cosine similarity scores calculated from Term Frequency Inverse Document Frequency (TF-IDF) weights [21]. After cleaning the textual content of each document, document vectors were calculated to enable their representation in a machine-readable format.

Previous document sources were fused together in one knowledge graph, which included the document vector representation, to enable the intelligent IR algorithm querying all types of documents uniformly. The construction of the knowledge graph was conducted following the steps in Section 3.1.

### 4.1.2. Knowledge Graph Construction

Based on the pillars for defining graph nodes and relations in our framework, we analyzed the three data sources in this use case to determine the domain requirements. We also interviewed the engineers to extract the expert-defined rules for constructing the graph. We started by modeling the three documentation sources using three node types in a multidimensional KG, see Figure 4. Each node type has properties that reflect the special content of the corresponding document it represents. Nodes that represent failure cases include the document vectors. The data content of these nodes was also arranged to represent the table columns in the original database. This table-like format

allows for the separation of each failure report into its main parts: "Failure Description", "Failure Tests", and "Failure Solution". Those parts are then used to enhance the similarity calculation with other nodes. Nodes that represent the chip specifications include the final vectors of the documents' content. Nodes that represent the project's structure included the textual vectors of the documents' content, alongside the structural information of each project element. This information includes the exact position of the project's element in the hierarchy of the designed chip.

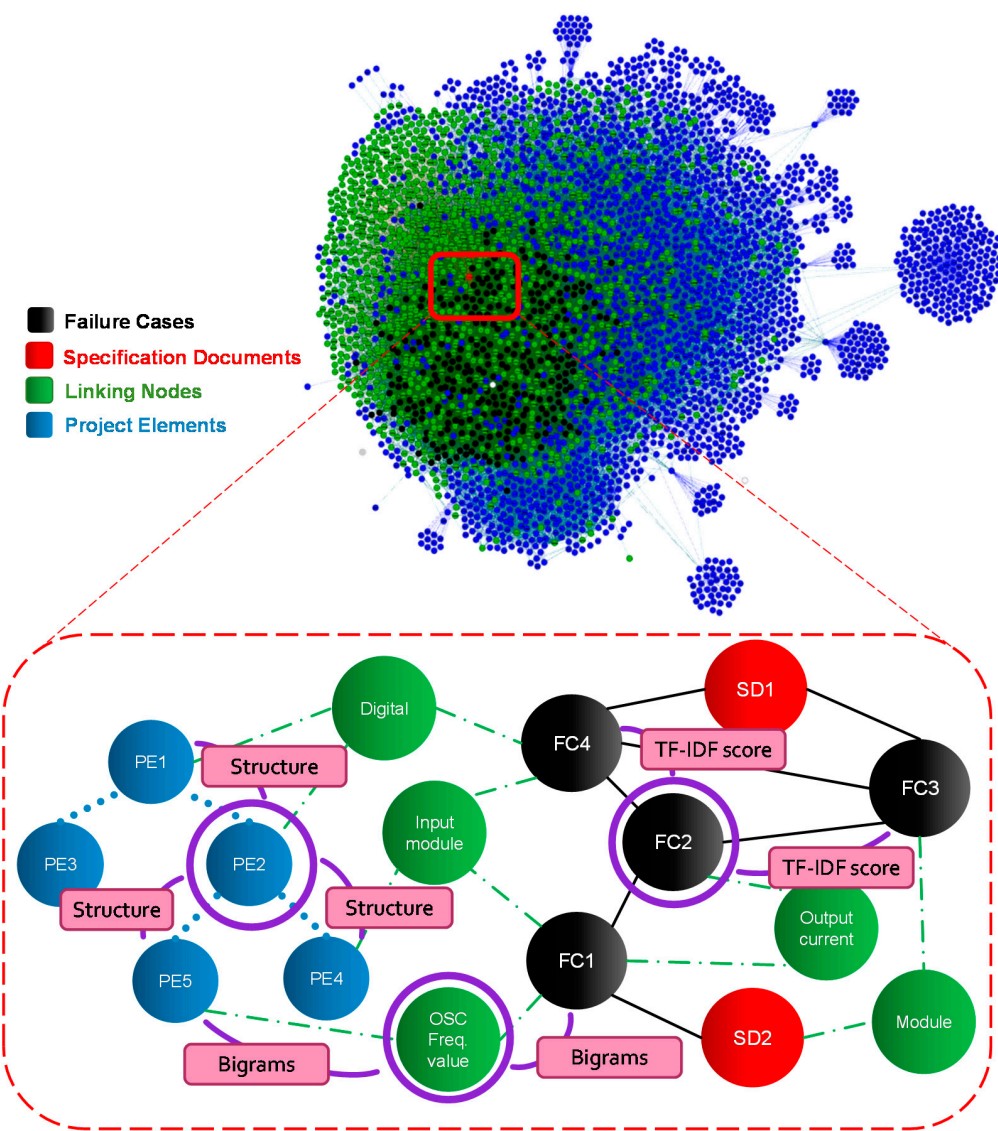

**Figure 4.** The construction of the multidimensional knowledge graph from chip-design documents. Four types of nodes are defined, along with their relations. Relations exist amongst nodes of the same type and among those of different types.

Expert defined rules led to the definition of a fourth type of nodes. We call the additional node type 'Linking nodes', and we use it to represent special documentation terms followed by the design engineers [8]. Such documentation style included, for example, the use of a special combination of terms as a shortcut of a common design problem. To capture those documentation styles, we extract different combinations of terms from the textual content. Those combinations form N-Gram terms, which include an N number of terms that appear in a specific repeating pattern amongst the documents. We use Unigrams, Bigrams,

and Trigrams, corresponding to patterns of 1-term, 2-terms, and 3-terms respectively. Those patterns are then used to enrich the connections in the knowledge graph.

Relations in the knowledge graph were extracted from the information sources, the expert defined rules, and the explicit domain requirements. From the information sources, textual similarity was calculated between the content of the three types of documents to define their relevance. To calculate the textual content similarity, we first implement a simple TF-IDF approach to get the term weights and generate the document vector. Using the calculated vectors, we use the Cosine-Similarity algorithm to find the relevant documents and then create relations between them in the KG. Expert defined rules and the corresponding node type 'linking nodes' were used to create extra relations between each node and the linking node's content, if it was included in the document's text. Project structural information was also a source of creating relations between the nodes, which corresponded to their hierarchical structure in the designed chip. Through these types of relations, documents from multiple sources were also transitively linked to each other, as shown in Figure 4.

### 4.1.3. Modeling Domain-Specific Features

In this use case, the chip-design domain introduced several requirements that the semantic search tool needed to consider. The first requirement is modeling different document types in one queryable knowledge base. The multidimensional knowledge graph we use in the framework accomplishes this task. The second requirement came from the documentation styles that the design engineers followed. This included the non-standard naming conventions, report structures, and the abbreviation types used in the documents. We model those requirements in our approach using the "domain requirements" and "expert-defined rules" components.

Modelling the domain-specific features influenced two parts of the system: (1) relation extraction and node definition in the KG, where special relations and nodes were defined for non-standard vocabulary used in the documents; and (2) the IR algorithm, which was modified in our implementation to handle free-text search-queries if they are composed solely of non-standard terms. Experts have provided a set of rules that guided the information retrieval and the explanation. Those rules included, for example, the importance of certain groups of failures over others. Such difference in the failure importance was a result of multiple factors, including the effect of such failure on other parts of the chip and the effect of the failure on the overall performance of the final product. This type of information was directly related to the expert's observations of the failure's long-term effects. Therefore, this knowledge was embedded in the KG and consequently the IR algorithm to put more emphasis on this type of failure. We implicitly design the search functions in the IR module to consider the set of expert-defined rules. This influences the search function to assign higher priority to specific documents within the search results and thus show them on top of the result list. The information that included the textual explanation was also defined by the domain experts and was modeled in the Explanation Requirement components in the framework to represent the expert's need for certain information in explaining the results of the IR algorithm.

### 4.1.4. IR Algorithm

For the defined use case, we implement a graph-based IR algorithm. The algorithm depends on the transitive relations in the knowledge graph to retrieve relevant documents to a user query, based on the shortest path approach. User queries are searched in the KG, revealing direct hits from graph nodes that correspond syntactically to the search terms. From the direct hits, relations in the KG are used then to retrieve new nodes that are connected to the direct-hit nodes. Several graph transitions can take place between the direct hit and the relevant result node in the transitive path. However, a high number of transitive relations will produce a large number of search results, with furthermore potentially lower relevance to the search query. Therefore, we set a threshold of 2 graph

transitions to retrieve relevant results. All retrieved results are then enriched with their corresponding explanations from the explanation algorithm and provided to the user through a search interface.

### 4.1.5. Explanation Algorithm

The explanation input included the rules that expert users provided in addition to the requirements that shape the explanation. In our use case, explanations are intended to clarify the relevance of a previous design-failure document to a user query. This explanation is generated from the semantic similarities between the documents and the properties of the document node. To represent those similarities in a human readable format, we design sentence templates that are built from the available information about the documents and their similarity. Pre-defined slots in the sentence template are filled with corresponding information to generate a full description of the algorithm's reasoning, which is shown in Figure 5. We define two templates with increasing quality of explanation. This is to consider the cases where there is a lack of information for filling the template's slots. This way, simple explanations with low quality will be generated in cases where no sufficient information is available, while more complex and informative explanations will be generated when more information can be collected from the KG. We refer to the latter type as high-quality explanations.

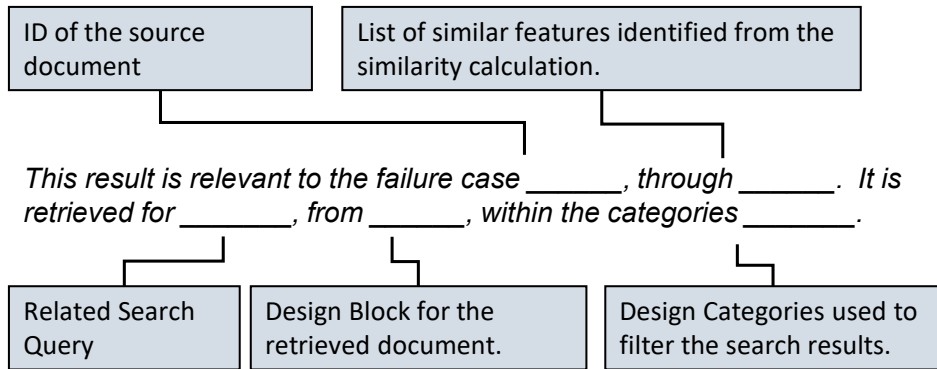

**Figure 5.** Verbal explanation template used in the implementation of Use Case 1. The template is designed based on expert requirements for explaining the semantic search results.

### 4.1.6. Evaluation and Results

We evaluate our framework in this use case through the metrics defined in Section 4. The components of the framework are evaluated in terms of their ability to represent the domain and generate meaningful, transferrable explanations and search results from the KG.

To evaluate how the proposed approach is capable of representing the domain of application, we use the Class Richness metric, see form (5). Here, we consider the entities in the framework as classes. We neglect the algorithm boxes since they do not follow a standard ontological representation. By defining the instances of each component in the use case and calculating the overall Class Richness of the knowledge graph, we achieve a value of 88% class coverage. This value reflects a high coverage rate of the proposed framework over the domain-specific requirements and their corresponding components in this use case.

To evaluate the IR and explainability algorithms, we perform a sample of 300 random searches through the IR algorithm. We firstly calculate the Relevance score (2) for each retrieved search result to evaluate the performance of the IR algorithm. Then, we calculate the Information Availability Equation (6), BLEU score, and Rouge-L score to measure the quality of the explanation generated for each result, Figures 6 and 7.

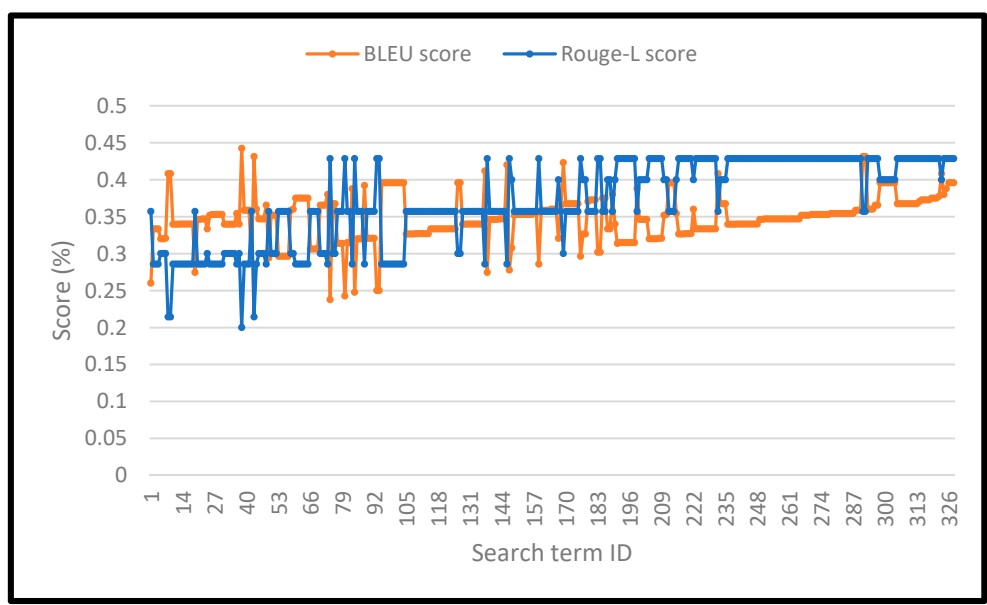

**Figure 6.** BLEU and Rouge-L scores for the search results. The figure represents the scores for the short search queries. The average BLEU score for the short search queries (on the horizontal axis) reached 37.2%, and the average Rouge-L score was 25.1%.

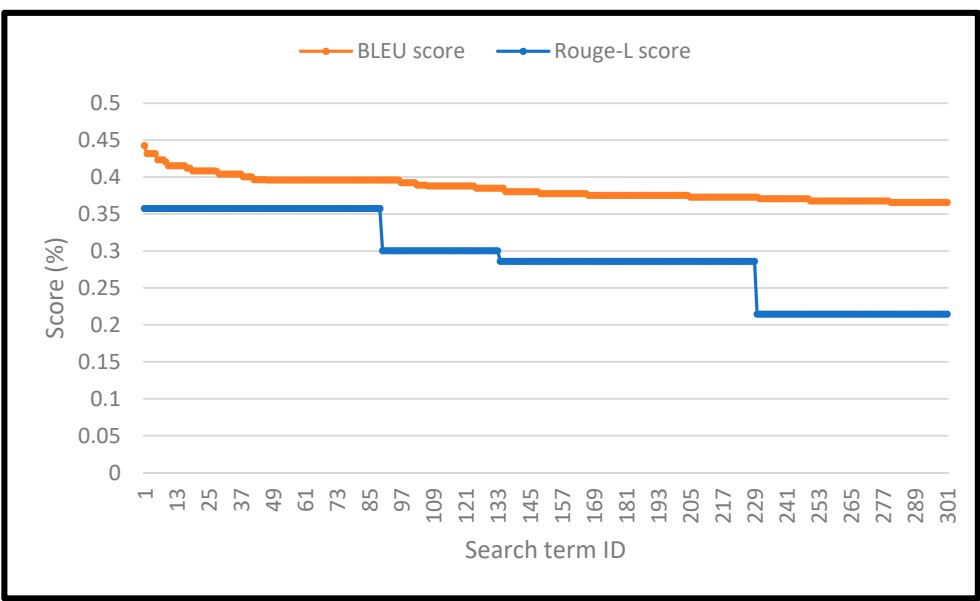

**Figure 7.** BLEU and Rouge-L scores for the search results. The figure shows the scores of the long search queries. The average BLEU score for the long search queries (on the horizontal axis) reached 38.3%, and the average Rouge-L score was 27.4%.

For each search query, we calculate the MEP score for the overall explanations generated for all retrieved results, as given in Equation (7). We achieve an average value of 99% for the MEP score. We trace this result back to the inherently open-box nature of the KG, which provides the IR and explanation algorithms with sufficient information about the retrieved results. Evaluation scores are shown in Table 2.

**Table 2.** Evaluation results of the proposed framework on Use Case 1.

| | Graph Evaluation Measure | IR Evaluation Measure | Explainability Evaluation Measure | | | |
|---|---|---|---|---|---|---|
| | Class Richness | Avg. Relevance | Information Availability | MEP | Avg. BLEU | Avg. Rouge-L |
| Short search query | 88% | 87.3% | 78.7% | 82.5% | 37.2% | 25.1% |
| Long search query | | 99% | 91.8% | 99.6% | 38.3% | 27.4% |

*4.2. Use Case 2: CV-Job Matching and Recommendation*

In the second use case, we choose a different domain to test the framework. We build an information retrieval algorithm that supports a job recommendation system. The field of matching job postings to job seekers features a different set of requirements, which is a result of the varying structures and content of job postings and job seekers' CVs [36].

CV documents include information that summarizes the user profile. Recommending a certain job to the CV holder requires finding the similarities between their skills and qualifications to those required by the job posting. This, in turn, requires the information retrieval algorithm to focus on parts of the CV that include the information about the user's qualifications and skills, more than other sections, such as the hobbies and interests. This also applies to the job postings themselves, where one can find several parts about the company portfolio, which are repeated in every job posting and thus do not participate in generating an effective matching to the specific job seeker.

To accomplish this matching effectively, we build a customized named entity classifier (NER) to support the information retrieval, and later the recommendation and explanation systems, in separating the different textual parts of the job posting and the user's CV to better model them in the knowledge graph.

4.2.1. Dataset Preparation

For this use case, we built an experimental dataset that included 50 user profiles and 1606 job postings. We collected the job descriptions in the fields of IT and Engineering from the job-hunting website "Monster" (https://www.monsterindia.com, accessed on 4 August 2020), in compliance with the European General Data Protection Regulations (GDPR) (https://gdpr-info.eu/art-5-gdpr, accessed on 10 December 2021). All user profiles were anonymized before being included in the dataset.

In the experiment dataset, we removed personal and demographic information about the user, without compromising the performance of the job recommendation system. This is because the recommender is designed as a graph-based collaborative-filtering (CF) system, following the shortest path algorithm of Dijkstra [30].

4.2.2. Knowledge Graph Construction

The knowledge graph is constructed from the two selected types of documents. It models them in two node types: userProfile (CV) nodes and jobPosting (JOB) ones. Node properties in this case include the textual content of multiple parts of the document, as well as the classified entities in the text, as predicted by the customized entity classifier.

The graph includes three types of relations:

(1)  userProfile-jobPosting which relates the user to potential jobs directly, see Figure 8.
(2)  userProfile-userProfile which finds the relevancies amongst users.
(3)  job-Posting-jobPosting relations.

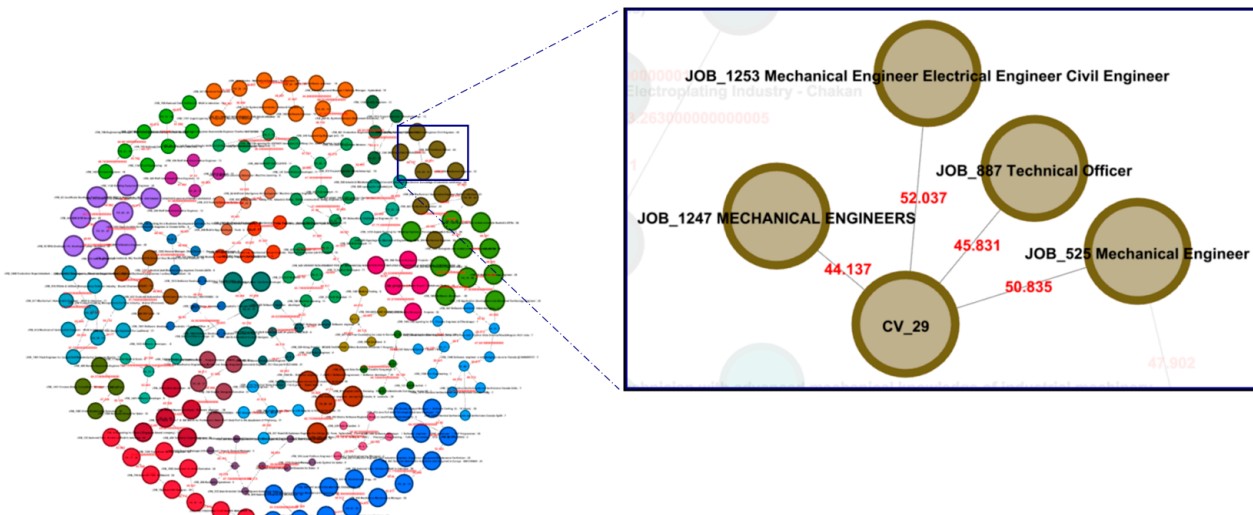

**Figure 8.** Direct relations between user profiles and relevant job postings. Each relation is defined based on the similarity score between the two nodes.

The last two relation types are extracted to support the CF-based recommender system. This happens through finding job posting recommendations based on the transitive relations in the knowledge graph. These create paths in the graph connecting a userProfile node to a jobPosting one, through an intermediate node, which is either a userProfile or a jobPosting, as shown in Figure 9.

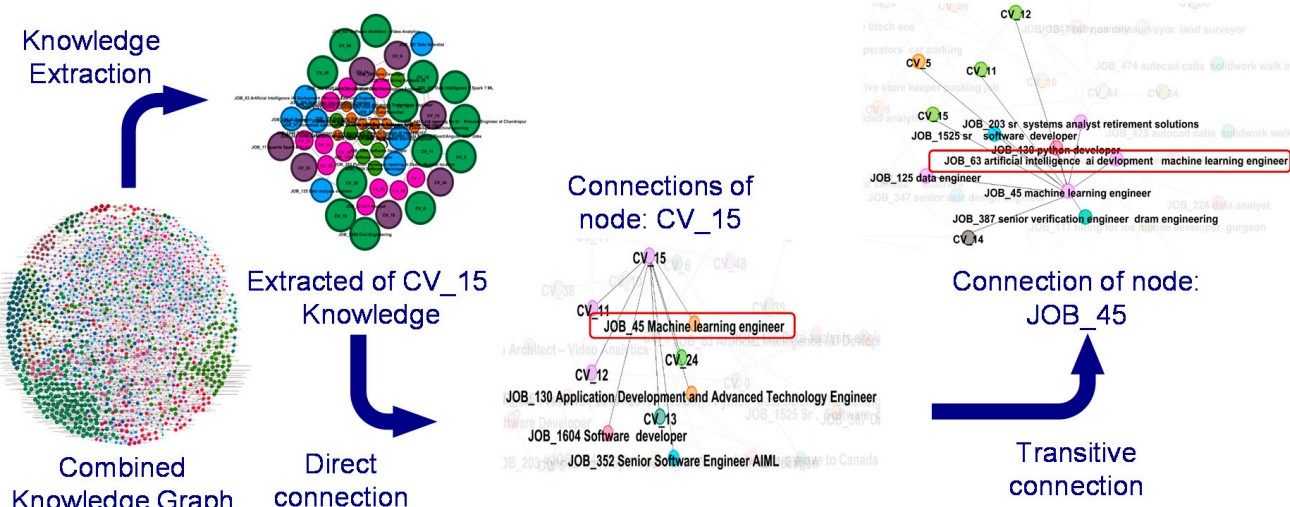

**Figure 9.** The use of direct and transitive relations in the knowledge graph. Here, jobPosting 63 is recommended to the userProfile_15, since it is relevant to the directly connected jobPosting_45.

Figure 10 shows the calculated similarities for two user profiles with the job postings in the database. From the similarity distribution analysis, we select the threshold value of >40% (Slightly above average distribution) and recommend the top 4 similar job postings to the respective user profile.

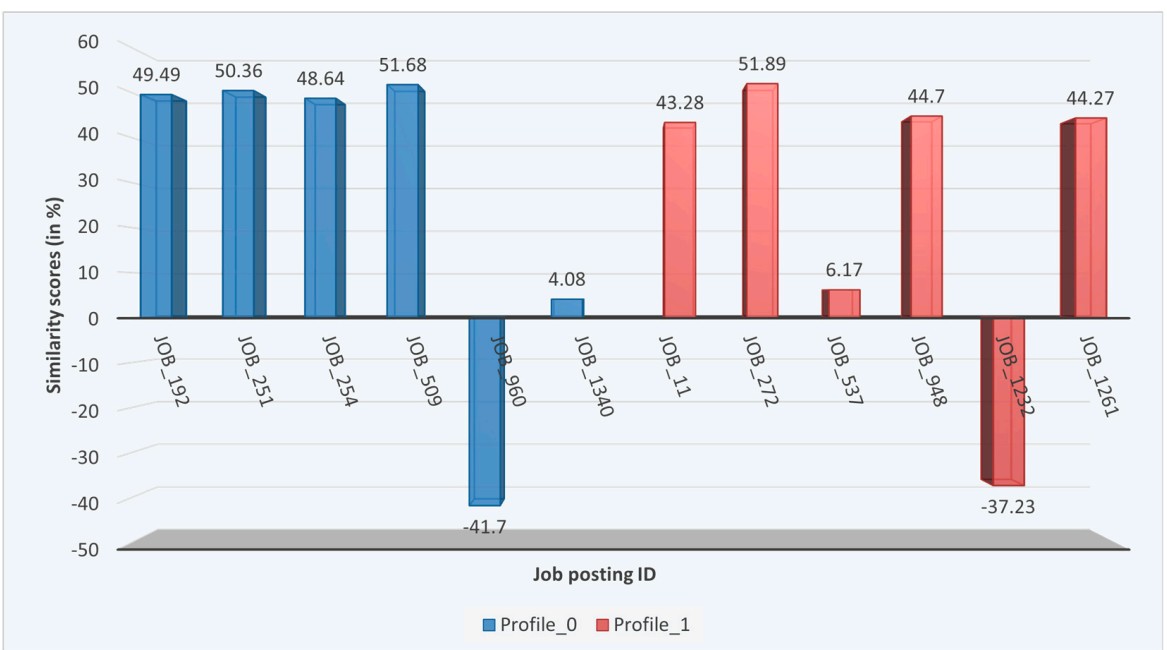

**Figure 10.** A sample of the overall calculated similarity scores between user profiles and job postings. Similarities above a predefined threshold are used to create corresponding relations in the knowledge graph. Negative similarity scores are caused by the angle value between document vectors, which results in a negative cosine value of that angle when calculating the cosine similarity.

### 4.2.3. Modeling Domain-Specific Features

The requirements of the job recommendation domain were very different from those in the chip-design one. The structure of user profiles and job descriptions, as well as the textual content in each one, revealed the need to classify the different sections of the document to enhance the recommendation result. This was a domain-specific requirement that we modeled through the "domain-requirements" component in the framework. It took the form of a list of document classes that created section-based relations in the knowledge graph and influenced the recommendations of a job postings.

Experts also provided rules on how different sections of the user profile have different levels of importance when generating the relation between a user profile node and a job posting one. Expert defined rules in this use case also influenced the textual content of document nodes. This, in turn, affected the document vectors and the corresponding similarities. An example of such rules would be the handling of numerical content of user profiles, where values of months and years were irrelevant to the relation extraction algorithm.

### 4.2.4. IR Algorithm and the Recommendation System

The information retrieval task in this use case is meant to provide the recommender system with needed information to generate a job recommendation for a user. The IR algorithm queries the KG as defined in the framework. Graph queries retrieve in this case all paths that link a certain userProfile node to neighboring jobPostings. Dijkstra's shortest path algorithm is used for this purpose.

The recommendations of relevant jobs are achieved on two levels:

(1) The direct level: where the user is recommended the most similar job postings that are directly connected to their corresponding node in the graph.
(2) CF-based recommendations that are recommended to the user based on the intermediate relevancy to another user or job posting. This means that a userProfile (A) can get a recommendation of a jobPosting (B) because userProfile (A) is directly connected to userProfile (B), which is directly connected to a jobPosting (B).

For example, if User (A) has a skill as a data scientist in their profile, while User (B) has a skill as a data analyst, then the similarity between those skills will link both users. In this case, User (A) will also receive a job recommendation for a Data Analyst position if it was recommended to User (B).

Similarly, another transitive path can go through an intermediate jobPosting node, as follows: userProfile (A) → jobPosting (A) → jobPosting (B). An example of this case would be: If User A receives a job recommendation for a Data Scientist position, and this position is already related to a Data Analyst position, then User (A) will also receive the recommendation for the Data Analyst position. An example of the path generation for the CF-based recommender is shown in Figure 9.

Figure 10 illustrates the cosine similarity scores calculated between two userProfil nodes and multiple jobPosting nodes in the KG. The horizontal axis represents the jobPosting in question, while the vertical axis represents the calculated similarity score. Each userProfile is illustrated in a different color. From Figure 10, it is noticeable that a similarity threshold is needed to determine the creation of a relation in the KG between the jobPosting and the userProfile. Negative scores that appear in Figure 10 are a result of the angle between the document vectors (here 90°). Those scores are considered in their absolute value in the graph relation definition.

### 4.2.5. Explanation Algorithm

The explanations generated for this use case used the same algorithm as in Use Case 1. Textual explanation templates were used with information slots, which are filled based on the knowledge graph. In this use case, however, the explanation requirements were different. This is because explaining the job recommendation needed to include specific information about the parts of the user profile that matched the job description. Those parts were categorized based on the custom entity classifier.

We design the NER as a support vector machine (SVM) and trained it on a custom dataset of 60,000+ entities. The dataset was created from documents in the technical and engineering domain. Those documents mainly represented job postings. The data was labeled manually by three independent technical personnel. Dataset labels have been chosen to correspond to the domain of job matching and recommendation. The classifier was therefore trained to predict 7 classes of entities that a user profile may include. These classes are programming_framework, skill, software, study, job_title, location, and programming_language. Based on the predicted classes, the similarity scores between a user profile and a job description, and the graph transitive relations, the explanation template was constructed as shown in Figure 11. The first explanation template uses syntactically similar words to fill the information slots with similar terminology between the user profile and the job posting. The second template is equipped with more information through implementing a customized name entity classifier on the language model. An example of the final explanation is:

"I recommend you Job number 509 as a "Software Engineer", which is a match to your profile by 51.682%, through the following classes:

Study: ['science', 'analysis', 'control', 'bachelor', 'data'],
Software: ['oracle'],
Programming language: ['java', 'python'],
Skills: ['agile', 'research', 'development', 'software']
Other similar terms to this job are: ['architecture', 'server', 'product', 'framework', 'design', 'technology']"

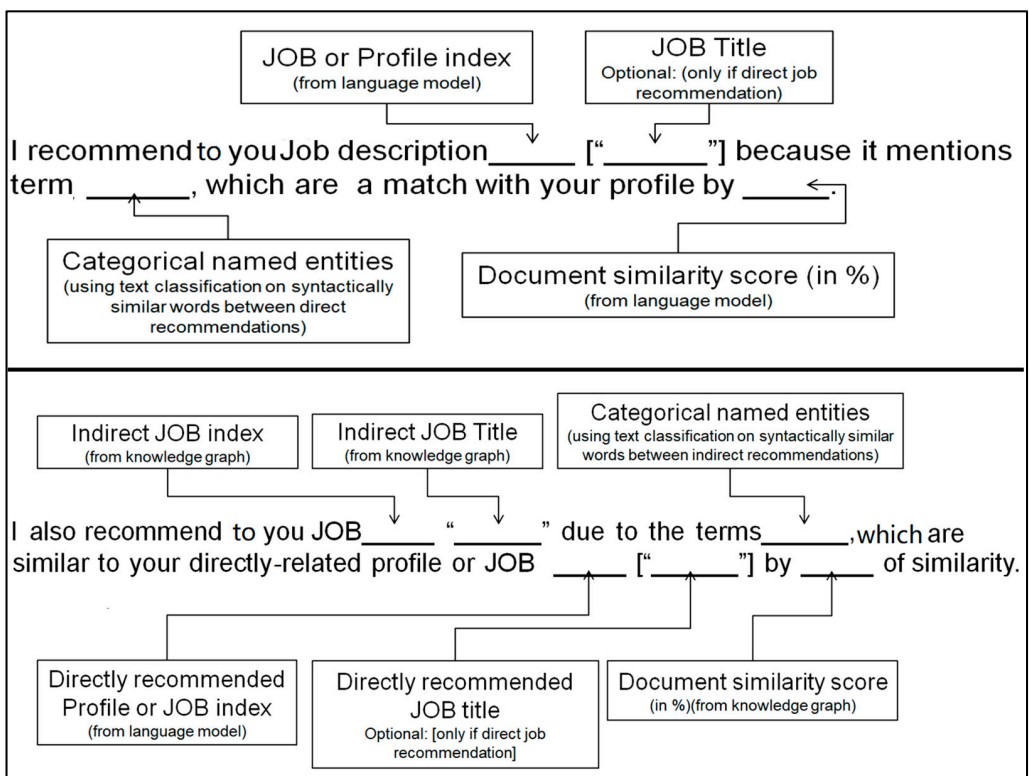

**Figure 11.** Two templates are used to generate the textual explanations. The first template is designed for direct recommendations, where only the link between the user profile and the job posting is explained. The second template includes information about the intermediate link between the user profile and the job posting, within the transitive relation.

### 4.2.6. Evaluation and Results

To evaluate the different parts of the framework in this use case, we calculate the KPI metrics we defined. Table 3 shows the accuracy, precision, and recall measures for the customized NER. We achieved an overall F1 score of 88% based on the training dataset in the domain of job matching.

**Table 3.** Precision, Recall, and F1 scores of the customized NER, with regard to the predefined classes.

|  | Precision | Recall | F1-Score | Support |
|---|---|---|---|---|
| programming_framework | 0.84 | 0.85 | 0.84 | 241 |
| skill | 0.794 | 0.88 | 0.83 | 2610 |
| software | 0.79 | 0.90 | 0.85 | 902 |
| study | 0.87 | 0.86 | 0.86 | 4481 |
| job_title | 0.89 | 0.99 | 0.94 | 480 |
| location | 0.99 | 0.96 | 0.98 | 485 |
| other_word | 0.93 | 0.88 | 0.90 | 9190 |
| programming_language | 0.92 | 0.98 | 0.895 | 642 |
| Accuracy |  |  | 0.88 | 19,031 |

To evaluate the knowledge graph structure, we use the Class Richness measure. We reach a percentage of 89% of coverage of the graph for the instances of CVs and job descriptions.

Evaluation of the explainability was conducted through calculating the MEP, BLEU, and Rouge-L scores for the explanations generated for recommended job postings. We achieve an MEP score of 85.6% for the recommended job postings. To calculate BLEU and Rouge-L scores, the user profile and its recommendations' content are combined for

reference, and the generated explanation is fed as a test sentence. The results are shown below in Figure 12 for multiple profiles and job postings. According to [37], the BLEU score and Rouge-L precision score are in a good range if they are above 20%, which shows that the generated explanations are related to the recommended job descriptions. Our BLEU score ranged around an average of 23.5%, and the Rouge-L scores ranged around an average of 26%.

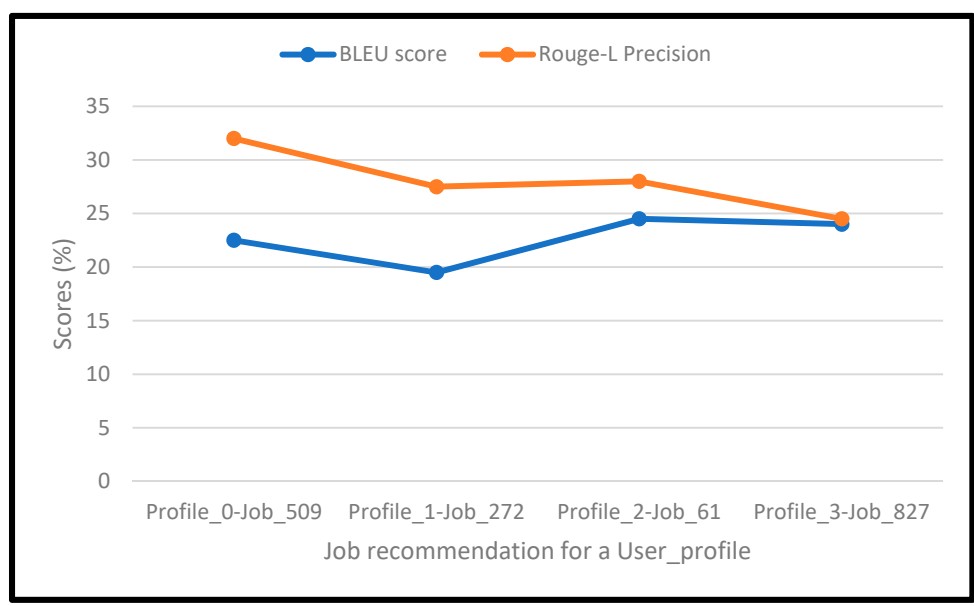

**Figure 12.** BLEU and Rouge-L scores for a sample of recommended job postings to two user profiles.

## 5. Discussion

The implementation of our framework in the presented use cases shows its adaptability to multiple domains while generating explanations for the retrieved search results and recommendations. In the evaluation experiments, we focused on the main contributions of our approach, namely the ability of the framework to embed domain-specific requirements and the transferability of the framework to multiple domains with minimum to no changes in the intelligent algorithms.

Embedding domain requirements in the two use cases was a direct result of adopting the domain-specific components in the proposed framework. Those requirements are captured from multiple parts of the system, including the rules that domain experts define, the results of the EDA process, and the content of databases. To the best of our knowledge, no other solution has been suggested in the literature as to how to include domain requirements from all of those parts. Here, we especially refer to the domain-specific solutions in the comparison Table 1, which only consider databases for extracting domain-related features. The importance of including other sources of domain requirements can be seen in the ability of expert defined rules in contrast to existing solutions, for example, to represent differences between the documents, which cannot be included in the database itself. This hinders other solutions from reaching the level of domain adaptability that the presented framework achieves.

Results from our experiments show that adopting the IR and explainability algorithms to requirements from a certain domain did not limit its ability to be transferred to other domains. The domains, within which we implemented our framework, revealed different features and requirements. Despite that fact, our use of the same IR and explainability algorithms was possible without compromising their accuracy. Numerical evaluation showed that the IR algorithm maintained an accuracy level higher than 95% in both use cases. The comparable literature in Table 1 focuses on the accuracy of the IR or recommendation algorithms, namely [11,15,24,25]. Within this context, we show in Tables 2

and 3 that our framework was able to achieve a high performance of the IR algorithm, ranging between 87% and 99% in both use cases, depending on the length of the user query and database size. This result is not only comparable to the 88.1% of accuracy, achieved by Wang et al. [15] using their Knowledge-aware Path Recurrent Network (KPRN), but adds to it the ability of considering domain requirements in the intelligent IR algorithm. Unlike the domain-intrinsic nature of the KPRN, our framework provides the same or higher accuracy in domain restricted environments, such as the ones demonstrated in the previous two use cases.

The explainability algorithm reached 85.6% in one use case and ranged between 82.5% and 99.6% depending on the length of the search query. In both cases the *MEP* score is comparable or surpasses the results of Chen and Miyazaki [17], who achieve an 87.25% *MEP* score from a similar explainability algorithm implemented on a Slack-Item-KG of 100k items. It should be also noted in this comparison that our main contribution is not achieving the highest accuracy levels of the IR and explainability algorithms but rather achieving accurate results in a domain-specific environment, with a higher level of restrictions and data complexity than in the general-purpose datasets used in [15,17,24,25]. The ability of our solution to consider those requirements and achieve high accuracy scores of the IR and explainability at the same time, is the value-added that our framework offers in comparison to the current state of the art. In Table 4, we summarize the comparative *MEP* scores of our proposed framework and other similar solutions.

**Table 4.** Explainability comparison based on the *MEP* scores.

| Approach | MEP Score % |
|---|---|
| Chen and Miyazaki—Slack-Item-KG 100k | 87.25 |
| Chen and Miyazaki—Slack-Item-KG 1M | 91.67 |
| Our framework—Use case1 (short query) | 82.5 |
| Our framework—Use case1 (long query) | 99.6 |
| Our framework—Use case2 | 85.6 |

It can be noticed from Table 4 that the MEP score increases with the increase of data size. This increase can be in the dataset itself, such as the case of using Slack-Item-KG 1M instead of 100k, or in the search query, such as the difference in our first use case between long and short queries. The enhanced potential for generating explanations from larger numbers of data is due to the increased potential to find relations and paths in the KG, which correspond to the user query. With more information in the search query, or larger knowledge graphs, more paths can be extracted and used for constructing the result explanation. Here, we mainly compared the explainability algorithm in our approach to the work of Chen and Miyazaki, since the authors explicitly use the MEP measure to evaluate their work. We then extended our evaluation to include BLEU and Rouge-L scores, as illustrated in Figures 6, 7 and 12.

The evaluation of the proposed framework was implemented on two use cases that have different domain requirements, to demonstrate the transferability of the solution. In general, the proposed framework is designed in a generic way that adopts to any domain of interest. The implementation of our framework in other domains, such as medical [23], energy [38], or educational [39], can be similarly accomplished, as long as the domain requirements are defined by the experts and data sources. The IR and explainability algorithms are then transferrable amongst those domains, due to their dependence on the KG structure. This fact implies the sustainability of the proposed framework, when used in a certain domain and then transferred to another one. Another sustainability aspect of our framework comes from the potential to develop and extend the knowledge graph itself, during the lifetime and therefore ongoing extension of the industrial process that is supported by the framework and its underlying data. Since the proposed framework includes the necessary elements to construct the knowledge graph, it inherently enables its

expansion and development. This allows for a long-term use of the framework, which can take into account new data, new domain requirements, and new user queries.

Our framework was developed to handle textual data sources. This fact influenced the nature of domain-specific and domain-agnostic components we used in the framework. That being said, we also address the limitation of our framework to include image data sources, since it lacks the image-processing components on the input part of the framework. Although the current framework excels in handling pure textual data sources, and imagery data that is well annotated, it can still be extended to include specific elements that handle pure visual data sources to embed them in the knowledge graph construction process. Once the image data is embedded in the KG, the IR and explainability algorithms will be able to query these data types directly, corresponding to the same user queries.

## 6. Conclusions

In this article, we proposed a framework for a domain-agnostic, explainable information retrieval based on knowledge graphs. Our framework is designed to model domain-specific requirements and make them available to the IR and explainability algorithms. We designed specialized components that are capable of embedding multiple data sources, expert-defined rules, and domain requirements in the construction of the KG. We use the KG as the center of our framework to tailor IR and explainability algorithms to the needs of a certain domain, without compromising their accuracy or transferability to other domains of application. The ability to use the same components of our proposed approach in different domains makes the overall solution domain agnostic. The architecture of the proposed framework enables the overall concept of (1) integrating domain-specific features in the IR algorithms and (2) enabling the transfer of IR and explanation functions between multiple domains.

We implemented and evaluated the proposed explainable approach in two real-life use cases, within the semiconductor chip design and the job recommendation domains. The implementation use cases showed the ability of our approach to represent multiple data sources and embed the expert knowledge effectively in the KG construction, the IR, and the explainability algorithms. We evaluated the framework based on three criteria: (1) its ability to cover the domain-specific requirements, (2) its transferability to other domains using the same sore components, and (3) the performance of information retrieval and the quality of the generated explanations. Evaluation results showed a high coverage of domain requirements that reached 88% in both evaluation domains. The same IR and explanation algorithms were used in both domains without changes, reflecting the transferability of the solution. Moreover, our framework implementation achieved MEP scores up to 99.6%, exceeding the comparable state of the art, and corresponding at the same time to the domains of interest. We measured the availability of information for generating the domain-specific explanations, which reached up to 91.8% in the case of long user queries. The performance of the IR algorithm was also tested and evaluated in terms of the relatedness of retrieved results to the user query. This relevance ranged between 87% and 99% depending on the length of the query. When using the IR results for generating job recommendations, our recommendation algorithm achieved a score of 88% for the F1 measure. Our evaluation of the intelligent, explainable, IR, and recommendation performance proved that the framework's domain-specific components and the use of knowledge graphs enabled generating high-quality explanations for the retrieved search results and recommendations in each of the evaluation domains.

Future steps for the development of our framework include its implementation in combination with the multidimensional knowledge representation (MKR) framework [40]. MKR is a framework that utilizes text mining in combining results from different dimensional analysis in one knowledge representation. Its ability to enhance the knowledge representation from analysis results, such as named entity recognition, topic detection, and sentiment analysis, holds the potential to support our framework in generating feature-rich explanations for the IR and recommendation algorithms.

**Author Contributions:** Conceptualization, H.A.-R., C.W. and J.Z.; methodology, H.A.-R., C.W., J.Z., M.D. and M.F.; software, H.A.-R., C.W. and J.Z.; validation, H.A.-R., C.W., J.Z., M.D. and M.F.; formal analysis, H.A.-R., C.W., J.Z., M.D. and M.F.; investigation, H.A.-R., C.W. and J.Z.; resources, H.A.-R., C.W., J.Z. and M.D.; data curation, H.A.-R., C.W. and J.Z.; writing—original draft preparation, H.A-R., C.W., J.Z. and M.D.; writing—review and editing, H.A.-R., C.W., J.Z., M.D. and M.F.; visualization, H.A.-R., C.W., J.Z. and M.D.; supervision, H.A.-R., C.W., M.D. and M.F.; project administration, H.A.-R., C.W. and J.Z. All authors have read and agreed to the published version of the manuscript.

**Funding:** This work was partially supported by the EU project iDev40. iDev40 received funding from the ECSEL Joint Undertaking (JU) under grant agreement No 783163. The JU receives support from the European Union's Horizon 2020 research and innovation program and Austria, Germany, Belgium, Italy, Spain, Romania. Information and results set out in this publication are those of the authors and do not necessarily reflect the opinion of the ECSEL Joint Undertaking.

**Institutional Review Board Statement:** Not applicable.

**Informed Consent Statement:** Not applicable.

**Data Availability Statement:** Data used in the first use case includes confidential information. Therefore, it is not publicly available. Data used in the second use case is publicly available as elaborated in Section 4.2.1. This data can be found here: (https://www.monsterindia.com, accessed on 4 August 2020).

**Acknowledgments:** Authors acknowledge and thank Chirayu Upadhyay for his support with the recommender system in Use Case 2.

**Conflicts of Interest:** The authors declare no conflict of interest.

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
