# Peer review of "Transferrable Framework Based on Knowledge Graphs for Generating Explainable Results in Domain-Specific, Intelligent Information Retrieval"

_informatics, doi:10.3390/informatics9010006_

Round 1

Reviewer 1 Report

The paper proposes a portable framework for generating domain-specific explanations for domain-dependent intelligent systems. The disadvantage of solutions that exist today is that the explainability of these systems has rarely been explored in domain-specific environments. In addition to accommodating domain requirements within the explainable intelligent information retrieval, the transfer of the explainable information retrieval algorithm to other domains remains an open problem. The method proposed in the article uses knowledge graphs to model domain knowledge. Knowledge graphs provides a solid foundation for developing intelligent infrared solutions. Using the same knowledge graphs, the authors develop graph-based components to generate textual and visual explanations of the information received, taking into account the requirements of the domain and maintaining portability to other domain-specific environments through a structured approach. The use of knowledge graphs resulted in minimum-to-zero adjustments when creating explanations for several intelligent information retrieval algorithms in several areas. The proposed method has been tested in two different scenarios of use: in the production of semiconductors and in the selection of a vacancy for a candidate. The quantitative results show the ability of the proposed approach to generate high-level explanations for end users. In addition, the developed explanation components could be easily adapted to both industries without compromising the overall accuracy of the intelligent information retrieval algorithm. Also, the article conducted high-quality user research.

Despite the satisfactory results of the research, there are some shortcomings that need to be corrected.

  1. The aim of the work should be defined.
  2. The state-of-art approaches should be separated by authors outcome to highlight their own results.
  3. The proposed method should be described in details. Now it looks like a concept.
  4. It is should be indicated whether estimation and evaluation metrics and approaches are developed by authors or are they well-known.
  5. Figure 7 should be divided to separate ones for better interpretation.
  6. Figure 8 should be described in text. The scales and the values are unclear.
  7. Discussion section should be included, which should contain comparison of obtained numerical results with existing methods.
  8. Conclusions should describe also numerical results.

Authors have interesting results, but structure of the paper should be upgraded. In summarizing my comments, I recommend that the manuscript is accepted after major revision.

Author Response

Dear Reviewer No. 1, 

Thank you for your valuable comments and remarks. We have summarized the changes we applied based on your comments in the following table: 

Transferrable Framework Based on Knowledge Graphs for Generating Explainable Results in Domain-Specific, Intelligent, Information Retrieval,

informatics-1520761

Reviewer 1 comments

Author comments / modifications

The aim of the work should be defined.

The aim of the work has been defined and elaborated in multiple places in the article. Please see addition in abstract, as well as:

·        Section 01, lines 73-76.

·        Section 01, lines 90-97.

·        Section 01, lines 108-114.

·        Section 01, lines 165-174

The state-of-art approaches should be separated by authors outcome to highlight their own results.

We addressed this issue by modifying table 1 to compare approaches based on their features and outcomes. We further highlight our result in the text to compare them to the surveyed literature.

The proposed method should be described in details. Now it looks like a concept.

We added several parts to the article in multiple sections to clarify the implementation of the framework. We added a paragraph about the reasons for choosing the representation method of knowledge graphs in section 01, lines 123-144.

We also utilize the implementation of the presented use cases to demonstrate that the framework is a practical solution and not only a theoretical concept.

It is should be indicated whether estimation and evaluation metrics and approaches are developed by authors or are they well-known.

A paragraph has been added to section 4, lines 545-550, about the different metrics used from literature as well as the own developed ones.

Figure 7 should be divided to separate ones for better interpretation.

The figure has been divided into two figures and re-captioned accordingly. Due to dividing other figures as well, former Figure 7 is now Figures 8 and 9.

Figure 8 should be described in text. The scales and the values are unclear.

Figure 8 is now Figure 10, and an additional paragraph explaining the scales and values has been added. Please see lines 846-853.

Discussion section should be included, which should contain comparison of obtained numerical results with existing methods.

A discussion section has been added (section 5, lines 915-1000). We address the numerical results in the discussion section, and we compare our results to those in the state of the art.

Conclusions should describe also numerical results.

Numerical results were addressed in the conclusion (section 6, lines 1024-1032).

A more detailed discussion of the results, and their comparison to the state of the art, was presented in the new discussion section 5.

Visibility of changes: New/Modified content has been indicated in the Word file in "Track Changes" mode.

Reviewer 2 Report

In this paper the authors propose a framework for a domain-agnostic, explainable information retrieval based on knowledge graphs. A list of points that appears to deserve to be better clarified in the paper together with some suggestions follows.

  • More details are needed to clarify the structure of the knowledge graph (section 4.1.2).
  • The data set which is used for training, validating and testing the proposed ANFIS should be described in more detail.
  • In section 4.1.3, the author mentioned this sentence “Experts have provided a set of rules that guided the information retrieval and the explanation. Those rules included, for example, the importance of certain groups of failures over others, which led to assigning higher priority to those documents within the search results, and thus, showing them on top of the result list.” More justification should be furnished on this issue.
  • Some assumptions are stated in various sections. More justifications should be provided on these assumptions. Evaluation on how they will affect the results should be made.
  • The authors do not clearly clarify their contributions in abstract and conclusions section.
  • Please introduce discussions with other articles in your conclusions. Provides three sample articles related to information retrieval and graphs:
    • Wael Ahmad Alzoubi, “Dynamic Graph based Method for Mining Text Data”, WSEAS Transactions on Systems and Control, Volume 15, 2020, pp. 453-458.
    • Wael Ahmad Alzoubi, “An Improved Graph based Rules Mining Technique from Text”, Engineering World, Volume 2, 2020, pp. 76-81.
    • Chuck Easttom, Mo Adda, “The Creation of Network Intrusion Fingerprints by Graph Homomorphism”, WSEAS Transactions on Information Science and Applications, Volume 17, 2020, pp. 124-131.

Author Response

Dear Reviewer No. 2, 

Thank you for your valuable comments and remarks. We have summarized the changes we applied based on your comments in the following table: 

Transferrable Framework Based on Knowledge Graphs for Generating Explainable Results in Domain-Specific, Intelligent, Information Retrieval,

informatics-1520761

Reviewer 2 comments

Author comments / modifications

More details are needed to clarify the structure of the knowledge graph (section 4.1.2)

A more detailed description of the construction process of the knowledge graph was added to section 4.1.2.

The data set which is used for training, validating and testing the proposed ANFIS should be described in more detail.

More description of the data sets and their use was added to both use cases, in section 4.1.1, and 4.2.5 respectively.

In section 4.1.3, the author mentioned this sentence “Experts have provided a set of rules that guided the information retrieval and the explanation. Those rules included, for example, the importance of certain groups of failures over others, which led to assigning higher priority to those documents within the search results, and thus, showing them on top of the result list.” More justification should be furnished on this issue.

A clearer justification of the idea in this statement has been added to section 4.1.3.

Some assumptions are stated in various sections. More justifications should be provided on these assumptions. Evaluation on how they will affect the results should be made.

Concrete lines / sections are missing here to directly relate this comment to the text. We rechecked the paper to determine the assumptions mentioned, and clarified as much as possible any pints in the implementation of the framework within the two use cases.

The authors do not clearly clarify their contributions in abstract and conclusions section.

The contribution of the work has been defined and elaborated in multiple places in the article. Please see addition in abstract, the conclusion, as well as:

·        Section 01, lines 73-76.

·        Section 01, lines 90-97.

·        Section 01, lines 108-114.

·        Section 01, lines 165-174.

·        Section 5, the discussion and comparison to the state of the art.

Please introduce discussions with other articles in your conclusions. Provides three sample articles related to information retrieval and graphs:

* Wael Ahmad Alzoubi, “Dynamic Graph based Method for Mining Text Data”, WSEAS Transactions on Systems and Control, Volume 15, 2020, pp. 453-458.

* Wael Ahmad Alzoubi, “An Improved Graph based Rules Mining Technique from Text”, Engineering World, Volume 2, 2020, pp. 76-81.

* Chuck Easttom, Mo Adda, “The Creation of Network Intrusion Fingerprints by Graph Homomorphism”, WSEAS Transactions on Information Science and Applications, Volume 17, 2020, pp. 124-131.

A discussion section has been added (section 5, lines 915-1000). We address the numerical results in the discussion section, and we compare our results to those in the state of the art.

Suggested references of W. A. Alzoubi have been added to section 02, lines 222-231, within their corresponding parts.

With careful consideration, we found that the third suggested paper was not directly related to the transferability of explainable IR systems amongst multiple domains.

Visibility of changes: New/Modified content has been indicated in the Word file in "Track Changes" mode.

Reviewer 3 Report

This paper proposes knowledge graph network to generate visual and textual interpretation of the information retrieval results. Computational results demonstrates the proposed approach can be a feasible solution for multiple information retrieval tasks. Overall, this research is interesting but it needs major revision before being considered for publication. The major issues are as fillows:

First, the proposed approach has not been compared with other benchmark methods. The authors only provided computational outcomes from the proposed approach which is insufficient to prove this is the best approach.

Second, the discussion section is missing. The authors should discuss the pros and cons for the proposed approach. Also, the comparison against the other approaches should be provided.

Third, some graphs are just screen shots from Excel. The authors encouraged to re-generate the graphs to provide a better display to the readers. For instance, Fig 6 & Fig 10 should be improved.

Last, the authors are encouraged to discuss how the proposed approach can be applied in the Energy industry. For instance, the author can refer the two papers below and discuss how the proposed approach can be applied in the energy dataset:

  • Li, H., Deng, J., Feng, P., Pu, C., Arachchige, D. D., & Cheng, Q. (2021). Short-Term Nacelle Orientation Forecasting Using Bilinear Transformation and ICEEMDAN Framework. Frontiers in Energy Research, 697.

  • Li, H., Deng, J., Yuan, S., Feng, P., & Arachchige, D. D. Monitoring and Identifying Wind Turbine Generator Bearing Faults using Deep Belief Network and EWMA Control Charts. Frontiers in Energy Research, 770.

After the revisions, this manuscript has the strong potential for being considered for publication.

Author Response

Dear Reviewer No. 3, 

Thank you for your valuable comments and remarks. We have summarized the changes we applied based on your comments in the following table: 

Transferrable Framework Based on Knowledge Graphs for Generating Explainable Results in Domain-Specific, Intelligent, Information Retrieval,

informatics-1520761

Reviewer 3 comments

Author comments / modifications

First, the proposed approach has not been compared with other benchmark methods. The authors only provided computational outcomes from the proposed approach which is insufficient to prove this is the best approach.

A comparison to other approaches has been introduced in section 2, Table 1, including an added explanation.

Furthermore, the new discussion section 5 has been added, where the results are discussed in detail and compared numerically to other approaches in te state of the art.

Second, the discussion section is missing. The authors should discuss the pros and cons for the proposed approach. Also, the comparison against the other approaches should be provided.

A discussion section has been added (section 5, lines 915-1000). We address the numerical results in the discussion section, and we compare our results to those in the state of the art. We also discuss the pros and cons of the proposed framework.

Third, some graphs are just screen shots from Excel. The authors encouraged to re-generate the graphs to provide a better display to the readers. For instance, Fig 6 & Fig 10 should be improved.

All figures in the article have been revised and re-inserted with maximum resolution. Figures that are constructed in Excel are now inserted alongside their original data, to provide the maximum visibility.

The captions for the figures have also been modified to make the differences more understandable.

Moreover, additional explanation to some figures has been added in the text, to enhance their interpretation.

Last, the authors are encouraged to discuss how the proposed approach can be applied in the Energy industry. For instance, the author can refer the two papers below and discuss how the proposed approach can be applied in the energy dataset:

* Li, H., Deng, J., Feng, P., Pu, C., Arachchige, D. D., & Cheng, Q. (2021). Short-Term Nacelle Orientation Forecasting Using Bilinear Transformation and ICEEMDAN Framework. Frontiers in Energy Research, 697.

* Li, H., Deng, J., Yuan, S., Feng, P., & Arachchige, D. D. Monitoring and Identifying Wind Turbine Generator Bearing Faults using Deep Belief Network and EWMA Control Charts. Frontiers in Energy Research, 770.

We have revised the suggested literature and added a clarification on the potential of implementing our framework in multiple domains, including the energy domain. This clarification can be found in the discussion section (Section 5).

Visibility of changes: New/Modified content has been indicated in the Word file in "Track Changes" mode.

Reviewer 4 Report

In order to increase the soundness of the paper, the following aspects must be improved:

  • please explain more clearly the novelty of the paper
  • please explain more clearly what are the reasons for choosing to propose a method based on semantic knowledge graph
  • comparison with other methods must be added
  • please explain how can be sustain the possibility of applying the proposed method to other domains (based on the evaluated  experiments)

Author Response

Dear Reviewer No. 4, 

Thank you for your valuable comments and remarks. We have summarized the changes we applied based on your comments in the following table: 

Transferrable Framework Based on Knowledge Graphs for Generating Explainable Results in Domain-Specific, Intelligent, Information Retrieval,

informatics-1520761

Reviewer 4 comments

Author comments / modifications

Please explain more clearly the novelty of the paper

The contribution and novelty of the work has been defined and elaborated in multiple places in the article. Please see addition in abstract, the conclusion, as well as:

·        Section 01, lines 73-76.

·        Section 01, lines 90-97.

·        Section 01, lines 108-114.

·        Section 01, lines 165-174.

·        Section 5, the discussion and comparison to the state of the art.

Please explain more clearly what are the reasons for choosing to propose a method based on semantic knowledge graph.

We added a paragraph about the reasons for choosing the representation method of knowledge graphs in section 01, lines 123-144.

Comparison with other methods must be added.

We addressed this issue with the modifications to table 1, to compare approaches based on their features and outcomes.

Additional explanations have been added in section2 to compare our approach with existing ones.

Furthermore, a new discussion section (Section 5) has been added with detailed comparisons of our numerical results to other methods in the state of the art.

please explain how can be sustain the possibility of applying the proposed method to other domains (based on the evaluated experiments)

In the new discussion section (Section 5, lines 915-1000) we also addresses the sustainability of our framework and its application in other domains.

Visibility of changes: New/Modified content has been indicated in the Word file in "Track Changes" mode.

Round 2

Reviewer 1 Report

Thanks to the authors for considering reviewer comments. In my opinion, now paper can be published in current form.

Reviewer 2 Report

The paper can be published as it is.

Reviewer 3 Report

The manuscript has been substantially improved and can be considered for publication in the current form

Reviewer 4 Report

Since all my comments are addressed, I recommend to publish the paper.